# The dispensability of 14-3-3 proteins for the regulation of human cardiac sodium channel Na$_v$1.5

Oksana Iamshanova[1]*, Anne-Flore Hämmerli[1], Elise Ramaye[1], Arbresh Seljmani[1,2], Daniela Ross-Kaschitza[1], Noëlia Schärz[1,2], Maria Essers[1], Sabrina Guichard[1], Jean-Sébastien Rougier[1], Hugues Abriel[1]*

1 Faculty of Medicine, Institute of Biochemistry and Molecular Medicine, University of Bern, Bern, Switzerland, 2 medi—Center for Medical Education, Bern, Switzerland

* oksana.iamshanova@unibe.ch (OI); hugues.abriel@unibe.ch (HA)

## Abstract

### Background

14-3-3 proteins are ubiquitous proteins that play a role in cardiac physiology (e.g., metabolism, development, and cell cycle). Furthermore, 14-3-3 proteins were proposed to regulate the electrical function of the heart by interacting with several cardiac ion channels, including the voltage-gated sodium channel Na$_v$1.5. Given the many cardiac arrhythmias associated with Na$_v$1.5 dysfunction, understanding its regulation by the protein partners is crucial.

### Aims

In this study, we aimed to investigate the role of 14-3-3 proteins in the regulation of the human cardiac sodium channel Na$_v$1.5.

### Methods and results

Amongst the seven 14-3-3 isoforms, only 14-3-3η (encoded by *YWHAH* gene) weakly co-immunoprecipitated with Na$_v$1.5 when heterologously co-expressed in tsA201 cells. Total and cell surface expression of Na$_v$1.5 was however not modified by 14-3-3η overexpression or inhibition with difopein, and 14-3-3η did not affect physical interaction between Na$_v$1.5 α-α subunits. The current-voltage relationship and the amplitude of Na$_v$1.5-mediated sodium peak current density were also not changed.

### Conclusions

Our findings illustrate that the direct implication of 14-3-3 proteins in regulating Na$_v$1.5 is not evident in a transformed human kidney cell line tsA201.

**Data Availability Statement:** All relevant data are publicly available on Zenodo as follows: The data underlying Fig 1 are at doi.org/10.5281/zenodo.8132741. The data underlying Fig 2 are at doi.org/

10.5281/zenodo.8132761. The data underlying Fig 3 and Table 4 are at doi.org/10.5281/zenodo. 8132784. The data underlying Fig 4 and Table 5 are at doi.org/10.5281/zenodo.8132792. The data underlying Fig 5 are at doi.org/10.5281/zenodo. 8132794. The data underlying S1, S2, S3, S4, and S5 Figs are at doi.org/10.5281/zenodo.10303505.

**Funding:** This work was funded by the Swiss National Science Foundation (https://www.snf.ch/ en) [SNF 310030_184783] to H.A. The funders had no role in study design, data collection and analysis, decision to publish, or preparation of the manuscript.

**Competing interests:** The authors have declared that no competing interests exist.

## Introduction

In the heart, the voltage-gated sodium channel, $Na_v1.5$ (encoded by *SCN5A* gene), is responsible for the genesis and propagation of the action potential in the contractile myocardium. The function of $Na_v1.5$ largely relies on the interaction with its protein partners that regulate proper localization, turnover, and biophysical properties of the channel [1]. Dysfunction of $Na_v1.5$ is associated with various lethal diseases such as cardiac arrhythmias and cardiomyopathy. Therefore, understanding the relationship between $Na_v1.5$ and its regulatory protein partners may uncover novel therapeutic strategies for patients suffering from *SCN5A*-related diseases.

One of the protein partners of $Na_v1.5$ was reported to be 14-3-3 proteins, which are a family of conserved regulatory molecules expressed in all eukaryotic cells. 14-3-3 can bind target proteins, modulating protein-protein interactions and the activity of their ligands due to inducing structural rearrangements, masking or unmasking of the functional sites, and changing the ligand's intracellular localization [2]. *YWHAB, YWHAG, YWHAE, SFN, YWHAH, YWHAQ,* and *YWHAZ* genes encode the mammalian 14-3-3 isoforms β/α, γ, ε, σ, η, θ, and ζ/δ, respectively, where α and δ represent the phosphorylated forms. All 14-3-3 isoforms exhibit a high level of structural homology and consist of 9 antiparallel α-helices arranged in an L-shape [2,3]. Through their N-terminal region, 14-3-3 proteins form homo- and heterodimers [3]. Each ~30-kDa monomer binds to a variety of the target proteins at the consensus sequences R(S/X)XpSXP, RXXXpSXP, and pS/pTX1–2-COOH of the phosphorylated motifs [4]. The best-studied examples of the unphosphorylated targets are P*seudomonas aeruginosa* virulence factor exoenzyme S and R18, a peptide identified from a phage display library [5,6]. R18 is widely used as a potent antagonist of 14-3-3 protein/ligand interactions due to its high affinity with all 14-3-3 isoforms ($K_d$ = 80 nM) followed by occlusion of their ligand-binding groove [5]. Accordingly, dimer of R18, difopein, is also a competitive non-isoform specific inhibitor of 14-3-3 protein/ligand interactions [7].

14-3-3 proteins target a wide variety of ligands and regulate many cellular processes [2]. Accordingly, 14-3-3 proteins were shown to be involved in cardiac physiology, including cardiomyocyte development, cell cycle, and $Ca^{2+}$ signaling [8–11]. Importantly, 14-3-3 was proposed to play a role in cardiac electrophysiology due to their interactions with cardiac ion channels, exchangers, and pumps [4]. Specifically, the α-subunit of $Na_v1.5$ channel was shown to coimmunoprecipitate with 14-3-3 proteins in mouse heart tissue as well as in heterologous expression systems [12]. Although 14-3-3η, -θ, and -ζ were identified by yeast 2-hybrid screen to directly interact with $Na_v1.5$, only 14-3-3η was investigated for functional consequences on the channel activity [12]. In the presence of exogenous 14-3-3η, the density of sodium current and activation curve were unchanged, while the inactivation curve was negatively shifted [12]. Based on this data, computer simulations suggested proarrhythmic effects of 14-3-3η [12]. Nevertheless, transgenic mice expressing a cardiac-specific loss-of-function 14-3-3η mutant exhibited normal cardiac morphology, cardiomyocyte appearance, and ventricular systolic function under basal conditions, while nothing was reported about its cardiac electrical activity [13]. Interestingly, immunostaining analysis revealed colocalization between exogenous 14-3-3η and $Na_v1.5$ in the intercalated discs of adult rabbit cardiomyocytes, suggesting a role in clusterization [12]. In line with these findings, 14-3-3 proteins were reported to mediate the coupled gating of $Na_v1.5$ [14]. Although two 14-3-3 binding sites were identified in the intracellular loop between domains I and II of $Na_v1.5$ by co-immunoprecipitation analysis, it was unclear which isoform mediated the channel gating properties through its direct interaction [14]. Therefore, our goal was to investigate the direct role of all 14-3-3 isoforms on $Na_v1.5$ in a heterologous expression system. Specifically, we addressed their interaction, expression, trafficking, and sodium current.

# Materials and methods

## cDNA constructs and cloning

Template cDNA constructs and their derivatives are listed in Table 1.

To clone *SCN5A*-WT into NanoBiT® CMV Flexi BiBit ready vectors, SgfI and PmeI restriction sites were introduced, for which the primers were designed by the Flexi® Vector

**Table 1. Description of cDNA constructs used in this study.**

| cDNA construct | Transcript (Genbank) | Source | |
|---|---|---|---|
| pcDNA3.1 | Empty vector | V86020, Invitrogen | |
| pcDNA3.1-*SCN5A*-WT | NM_000335.5 | GenScript, NJ, USA | |
| pcDNA3.1-3X-FLAG-*SCN5A*-WT | | | |
| pcDNA3.1-HA-*SCN5A*-WT | | | |
| pFN217K-LgBiT(Nter)-*SCN5A*-WT | | Cloned from pcDNA3.1-*SCN5A*-WT into NanoBiT® CMV Flexi BiBit ready vectors (ID# CS1603B33) according to Flexi® vector systems technical manual #TM254 from Promega (Switzerland) | |
| pFC219K-*SCN5A*-WT-LgBiT(Cter) | | | |
| pFN218K-SmBiT(Nter)-*SCN5A*-WT | | | |
| pFC220K-*SCN5A*-WT-SmBiT(Cter) | | | |
| CMV/SmBiT-CA/BlastR | NM_002730.4 | NanoBiT® CMV PPI control vectors (ID# CS1603B54) from Promega (Switzerland) | |
| CMV/LgBiT-R2A/HygR | NM_004157.4 | | |
| BiBiT-RI/SmBiT-CA/LgBiT-R2A/BlastR | NM_002730.4 and NM_004157.4 | | |
| CMV/HaloTag®-SmBiT | HM157289.1 | | |
| pcDNA3-HA-*YWHAB*-WT | NM_003404.5 | a gift from Michael Yaffe [15] | Addgene #13270 |
| pcDNA3-HA-*YWHAG*-WT | NM_012479.4 | | Addgene #13274 |
| pcDNA3-HA-*YWHAE*-WT | NM_006761.5 | | Addgene #13273 |
| pcDNA3-HA-*SFN*-WT | NM_006142.5 | | Addgene #11946 |
| pcDNA3-HA-*YWHAH*-WT | NM_003405.4 | GenScript, NJ, USA | |
| pcDNA3-HA-*YWHAQ*-WT | NM_006826.4 | | |
| pcDNA3-HA-*YWHAZ*-WT | NM_003406.4 | | |
| pcDNA3-FLAG-*YWHAB*-WT | NM_003404.5 | a gift from Dario Diviani (University of Lausanne, Switzerland) | |
| pcDNA3-FLAG-*YWHAZ*-WT | NM_003406.4 | | |
| pSCM YFP-difopein | Peptide sequence: SADGAPHCVPRDLSWLDLEANMCLPGAAGLDSADGAPHCVPRDLSWLDLEANMCLPGAAGLE | a gift from Haian Fu (Emory University School of Medicine, GA, USA) | |
| pSCM YFP-mutant difopein R18(D12K,E14K) | Peptide sequence: PHCVPRDLSWLKLKANMCLP | | |
| pIRES-*CD8*-WT | NM_001768.7 | a gift from David Kass (Johns Hopkins University, MD, USA) | |
| pIRES-*CD8*-WT-*SCN1B*-WT | NM_001768.7 and NM_001037.5 | | |

Primer Design Tool (https://ch.promega.com/techserv/tools/flexivectortool/) and purchased from Eurofins Genomics (Germany) (forward primer 5'-3': `GTC GGC GAT CGC CAT GGC AAA CTT CCT ATT ACC TCG G`; reverse primer 5'-3': `GCG AGT TTA AAC CAC GAT GGA CTC ACG GTC CC`).

## Cell culture and transfection

Human embryonic kidney cells tsA201 (ECACC Cat# 96121229, RRID:CVCL_2737) were cultured up to 20 passages at 37°C with 5% $CO_2$ in Dulbecco's modified Eagle's culture medium (41965, Gibco™, Thermo Fisher Scientific, USA), supplemented with 2 mM L-glutamine (G7513, Sigma), 50 U/mL penicillin-streptomycin (15140122, Gibco™) and 10% heat-inactivated fetal bovine serum (10270–106, Lot: 2440045, Gibco™). When needed, cells were split using phosphate-buffered saline (PBS, 10010–015, Gibco™) and 0.05% Trypsin-EDTA (25300–054, Gibco™). Mycoplasma contamination status was tested weekly with PCR Mycoplasma Test Kit I/C (PK-CA91-1096, Promokine, PromoCell GmbH, Germany). cDNA constructs were introduced into cells by transfection with LipoD293™ (SL100668, SignaGen® Laboratories). Cells were harvested 48 hours after transfection unless stated otherwise. Stable cell lines tsA201-*SCN5A*-WT and tsA201-FLAG-*SCN5A*-WT were established by polyclonal selection and were maintained in culture growth media with 200 μg/mL Zeocin™ Selection Reagent (R25005, Gibco™).

## RNA extraction, reverse transcription polymerase chain reaction, and agarose gel electrophoresis

Cell monolayers were once washed with PBS, detached with 0.05% Trypsin-EDTA, and pelleted by centrifugation at 200 rpm for 5 minutes. Afterwards, cell pellets were washed with PBS three times more and then taken for RNA extraction using ReliaPrep™ RNA cell Miniprep Systems according to the manufacturer's instruction (Z6010, Promega). The concentration of extracted RNA was determined by NanoDrop™ One/One$^C$ Microvolume UV-Vis Spectrophotometer (ND-ONE-W, Thermo Fischer Scientific™). Reverse transcription (RT) was performed using a High-Capacity cDNA Reverse Transcription kit (4375222, Applied Biosystems) followed by polymerase chain reaction (PCR) with GoTaq® G2 Master Mix (M7822, Promega) on Biometra TRIO Touch Thermocycler (207072X, LabGene Scientific, Switzerland). The PCR protocol was as follows: initial denaturation at 95°C for 2 minutes, 35 cycles of denaturation at 95°C for 1 minute, annealing at 60°C for 45 seconds and elongation at 72°C for 45 seconds per kb, and a final extension of 72°C for 5 minutes. PCR products were stored at 4°C. Primers with their expected amplicon sizes are listed in Table 2.

## Protein lysate extraction

Pre-washed (PBS) adherent monolayers of cells were scraped in 2 mL of cold PBS (pH 7,4) and pelleted by centrifugation for 5 minutes at 200 g at 4°C. Cell pellets were lysed in lysis buffer (50 mM NaCl, 50 mM imidazole/HCl, 2 mM 6-aminohexanoic acid, 1 mM EDTA, pH 7) with addition of cOmplete tablets EDTA-free (04693132001, Roche, Switzerland), 0.5 mM $Na_3VO_4$, 0.5 mM NaFl, 10 μg/mL aprotinin, 10 μg/mL leupeptin, 1 mM phenylmethylsulfonyl fluoride and 1% digitonin (D141, Sigma) for 1 hour at 4°C. Lysates were centrifuged at 16,100 g for 15 minutes at 4°C. The obtained supernatant was taken for further analysis. Protein concentration was determined by Coo Assay Protein Dosage Reagent (Uptima, UPF86420).

**Table 2. Description of primers used in this study.**

| Target gene | Forward primer 5'-3' | Reverse primer 5'-3' | Amplicon size, bp |
|---|---|---|---|
| GAPDH | CCTTCATTGACCTCAACTACATGG | GGGATCTCGCTCCTGGAAGATGG | 140 |
| PSMB4 | GCACTTTACAGAGGTCCAATCAC | CTCGGTAGTACAGCACTCGC | 565 |
| TBP | CTTGACCTAAAGACCATTGCACTTC | GAAACTTCACATCACAGCTCCC | 266 |
| YWHAB | CTGCTCTCTGTTGCCTACAAGAATGT | CTTCGTCTCCCTGGTTTTTCCGAT | 583 |
| YWHAG | GCGAGCAACTGGTGCAGAAAGC | GTACTGGGTCTCGCTGCAATTCTT | 341 |
| YWHAE | GACAAGCTAAAAATGATTCGGGAATATCG | CTGAAGTCCATAGTGTCAGATTATCACG | 472 |
| SFN | GAGAGAGCCAGTCTGATCCAGAA | GTAGTAGTCACCCTTCATCTTCAGGTAG | 381 |
| YWHAH | TACGACGACATGGCCTCCG | CAAACTGTCTCCAGCTCCTTCTCA | 233 |
| YWHAQ | GCTCTCCGTGGCCTACAAGAAC | CGATCATCACCACACGCAACTTCA | 285 |
| YWHAZ | GATGACAAGAAAGGGATTGTCGATCAGT | CACTTAATGTATCAAGTTCAGCAATGGCT | 217 |

## Co-immunoprecipitation

For immunoprecipitation nProtein™ A Sepharose 4 Fast Flow (17528001, Cytiva) or anti-FLAG® M2 Magnetic Beads (M8823, Sigma) were used in the proportion of 1 μL of beads per 20 μg of total protein lysate. Protein lysates were diluted in Tris-buffered saline (TBS, 20 M Tris, 0.1 mM NaCl, pH 7.6, HCl-adjusted) and were mixed with beads that were washed three times with TBS and then coupled with antibody at 4°C overnight. After washing the beads with TBS for six times, the co-immunoprecipitated protein complex was eluted with 4X LDS sample buffer (NP0007, Invitrogen). The samples were analyzed with the immunoblotting technique (see below).

## Cell surface biotinylation assay

Adherent cell monolayers were gently washed three times with cold PBS (pH 8) and incubated, while slowly rocking, with 1 mg/mL EZlink™ Sulfo-NHS-SS-Biotin (21331, Thermo Fisher Scientific) in PBS (pH 8) for 45 minutes at 4°C. Afterward, the excess biotin was quenched by washing with 50 mM Tris-HCl (pH 8), followed by several washes with PBS (pH 7,4). Extracted protein lysates were mixed with Streptavidin Sepharose High-Performance beads (GE Healthcare, USA) at 4°C overnight. After thoroughly washing the beads with washing buffer (0.1% BSA, 0.001% Tween20, PBS pH 7,4), the biotinylated fraction was eluted with 4X LDS sample buffer and analyzed with the immunoblotting technique (see below).

## Sodium dodecyl sulfate–polyacrylamide gel electrophoresis (SDS-PAGE) and immunoblotting analysis

Protein lysates containing LDS sample buffer and 100 mM dithiothreitol (DTT) were loaded onto 4–12% Tris-acetate acrylamide gels and let to run at 60 mV in Tris-acetate running buffer (50 mM Tris, 50 mM Tricine, 0.1% SDS, pH 8.3). Afterward, proteins were transferred to the nitrocellulose membrane by using the Trans-Blot Turbo system (1704158, Bio-Rad, USA). After validation of successful protein transfer with Ponceau S solution (0.1% Ponceau S, 12.5% acetic acid), the membranes were blocked with 5% bovine serum albumin (BSA) in TBS-T (TBS, 0.1% Tween20) for 1 hour at room temperature and then incubated with primary antibody dilutions at 4°C overnight. After incubating membranes with secondary antibodies for 1 hour at room temperature, the infrared fluorescent signals were revealed with LiCor Odyssey Infrared imaging system (LI-COR Biosciences) and quantified with ImageJ software (Rasband, W.S., U. S. National Institutes of Health, Bethesda, Maryland, USA). For the analysis of Na$_v$1.5 protein level, the predominant band around 205 kDa was quantified.

**Table 3. Description of primary and secondary antibodies used in this study.**

| Antigen | Dilution | Host | Epitope | RRID | Source |
|---------|----------|------|---------|------|--------|
| FLAG | 1:1000 | mouse | DYKDDDDK | AB_262044 | F1804, Sigma-Aldrich |
| GFP | 1:1000 | mouse | n/a | AB_390913 | 11814460001, Roche (Switzerland) |
| GFP | 1:1000 | rabbit | Full length denatured and non-denatured TurboGFP and CopGFP | n/a | AB514, Lot#51402250467, Evrogen |
| HA | 1:500 | mouse | YPYDVPDYA | n/a | ENZ-ABS118-0500, Enzo Life Sciences |
| HA | 1:1000 | rabbit | YPYDVPDYA | AB_2810986 | ab137838, Abcam (UK) |
| $Na_v1.5$ | 1:1000 | rabbit | DTVSRSSLEMSPLAPV | n/a | generated by Pineda (Germany) |
| 14-3-3η | 1:1000 | rabbit | n/a | n/a | ab206292, Abcam (UK) |
| $\alpha1$-$Na^+$/$K^+$ ATPase | 1:500 | mouse | Full length native $\alpha1$-$Na^+$/$K^+$ ATPase | AB_306023 | ab7671, Abcam (UK) |
| $\alpha1$-syntrophin | 1:1000 | rabbit | SGRRAPRTGLLELRAC | n/a | generated by Pineda (Germany) |
| $\alpha$-actin | 1:1000 | rabbit | SGPSIVHRKCF | AB_476693 | A2066, Sigma-Aldrich |
| IgG mouse | 1:20,000 | goat | Reacts with the heavy and light chains of mouse $IgG_1$, $IgG_{2a}$, $IgG_{2b}$, and $IgG_3$, and with the light chains of mouse IgM and IgA. | AB_10956588 | 926–68070, LI-COR Biosciences |
| IgG rabbit | 1:20,000 | goat | Reacts with the heavy and light chains of rabbit IgG, and with the light chains of rabbit IgM and IgA. | AB_621843 | 926–32211, LI-COR Biosciences |

Primary and secondary antibodies used in this study are listed in Table 3.

## Patch clamp electrophysiology

To test data replicability, whole-cell sodium current recordings from two researchers (one of whom was blinded) using different setups were pooled together: either with an Axopatch 200B amplifier (Molecular Devices, Wokingham, United Kingdom) or with a VE-2 amplifier (Alembic Instrument, USA). Cells, solutions, micropipettes, and protocols were identical.

All cells were transiently co-transfected with *CD8*. Only successfully transfected cells, visualized by brief pre-wash with 1:1000 dilution of Dynabeads™CD8 (Thermo Fisher Scientific), were chosen for electrophysiological recordings. Thin-wall capillaries (TW150F-3, World Precision Instruments, Germany) were pulled with a DMZ-universal puller (Zeitz, Germany) with a tip resistance of 1.0 to 3.0 MΩ. The intracellular pipette solution contained (in mM): CsAsp 70, CsCl 60, EGTA 11, $CaCl_2$ 1, $MgCl_2$ 1, HEPES 10, and 5 $Na_2ATP$. The extracellular solution contained (in mM): NaCl 20, NMDG-Cl 110, $CaCl_2$ 2, $MgCl_2$ 1.2, HEPES 10, CsCl 5, D-glucose 5. pH was adjusted to 7.2–7.4 with CsOH or HCl. Osmolarity was measured by Osmometer type OM 806 (Löser) in mOsmol/kg $H_2O$ range. All recordings were performed at a stable temperature of 25°C using Axon™ pCLAMP™ 10 Electrophysiology Data Acquisition & Analysis Software, Version 10.2 (Axon Instruments, CA, USA). The recorded sodium current $I_{Na}$ was filtered with a low-pass Bessel 5 kHz filter at a sampling rate of 20 kHz per signal. Recordings without leak subtraction were selected for analysis. Raw traces were not corrected for the liquid junction potential.

Exclusion criteria for individual traces were: absence of $I_{Na}$; current at holding potential (-100 mV) < = -100 pA; slope of the activation (A) curve < = 4; and slope of the steady-state inactivation (SSI) curve > = 8. Sodium current density (pA/pF) was calculated by dividing peak current by cell capacitance. Current-voltage (I–V) curves were fitted with the equation $y = g(V_m - V_{rev})/((1 + \exp[(V_m - V_{1/2})/k]))$, where $y$ is the normalized peak current (pA/pF) at a given holding potential, $g$ is the maximal conductance, $V_m$ is the membrane potential, $V_{rev}$ is

the reversal potential for Na$^+$, $V_{1/2}$ is the potential at which half of the channels are activated and $k$ is the slope factor of activation. Activation (A) curves were fitted with the Boltzmann equation $y = 1-(1/(1 + \exp[(V_m-V_{1/2})/k]))$, where $y$ is the normalized conductance at a given holding potential, $V_m$ is the membrane potential, $V_{1/2}$ is the potential at which half of the channels are activated and $k$ is the slope factor of activation. SSI curves were fitted with the Boltzmann equation $y = 1/(1 + \exp[(V_m-V_{1/2})/k])$, where $y$ is the normalized current at a given holding potential, $V_m$ is the membrane potential, $V_{1/2}$ is the potential at which half of the channels inactivated and $k$ is the slope factor of inactivation. Time constants of fast ($\tau_{fast}$) and slow inactivation ($\tau_{slow}$) were extracted from fitting onset of current decay during the step pulse with a two-component exponential according to $f(t) = \sum_{i=1}^{n}(A_i1 \cdot e -t/\tau_i)+C$. Data analysis was not blinded.

## Protein-protein interaction assay

The NanoLuc® Binary Technology, NanoBiT® (Promega) allowed us to monitor protein-protein interactions between two Nav1.5-WT subunits in live cells using an aqueous, cell-permeable furimazine substrate. For this purpose, *SCN5A*-WT was cloned into different vectors that encoded Nav1.5-WT tagged with Large BiT (LgBiT) and Small BiT (SmBiT) on N- or C-termini when transfected into tsA201 cells. In the case of physical interaction between LgBiT and SmBiT, the bright luminescent signal was formed. Since Nano-Glo® Live Cell Assay (N2012, Promega) was not ratiometric and depended on the cell quantity, we normalized the luminescent signal to the fluorescent signal obtained with CellTiter-Fluor® Cell Viability assay (G6082, Promega). The signals were detected with Spark® multimode microplate (Tecan) and GloMax® Explorer Multimode Microplate Reader (GM3500, Promega).

## Data and statistical analysis

Data are represented as means ± SEM. Data normality was tested by using Shapiro-Wilk test. Statistical significance for normally distributed data was calculated with ordinary one-way ANOVA and Tukey's multiple comparisons test. In case data were normalized to control condition statistical significance was calculated with one-sample two-tailed t-test with hypothetical mean value = 1. Statistical significance for non-normally distributed data was calculated with Kruskal-Wallis test (Prism version 8.4.3; GraphPad, CA, USA). Exact *p*-values are indicated in the figures.

## Results

### Identification of 14-3-3 isoforms interacting with Nav1.5

To identify which 14-3-3 isoform interacts with Nav1.5, we performed co-immunoprecipitation analysis. Individual overexpression of all seven human 14-3-3 isoforms tagged with HA revealed that by themselves 14-3-3 proteins have a certain degree of non-specific binding with the anti-FLAG magnetic beads (S1 and S2 Figs). Accordingly, the control condition with non-tagged Nav1.5 was run along with FLAG-Nav1.5 and compared for the intensity of co-immunoprecipitated 14-3-3 protein. As an indicator of successfully co-immunoprecipitated Nav1-5-specific fraction, we used α1-syntrophin, a known partner of Nav1.5 macromolecular complex and endogenously present in tsA201. From all the isoforms, only 14-3-3η exhibited reproducibly brighter band intensity in the FLAG-immunoprecipitated fraction compared to the non-tagged fraction; hence, we concluded that only this isoform reliably interacts with Nav1.5 (S1 and S2 Figs). Furthermore, sepharose beads coupled with anti-Nav1.5 antibody successfully co-immunoprecipitated α1-syntrophin and 14-3-3η, but not 14-3-3σ (Fig 1A). However, the

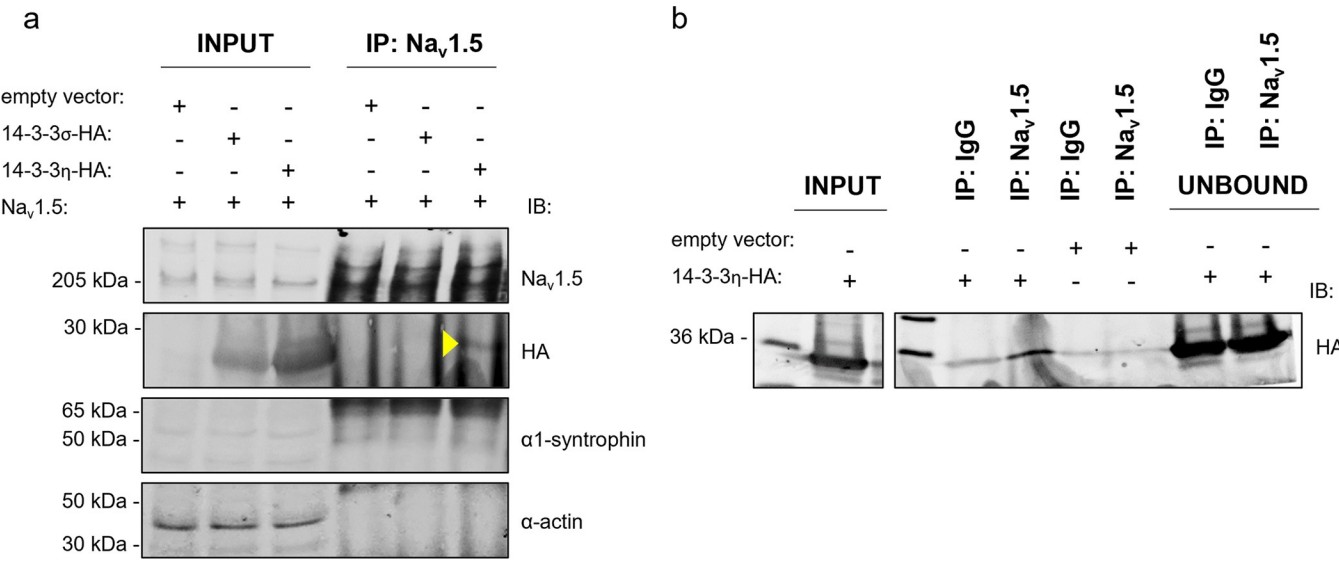

**Fig 1. Na$_v$1.5 co-immunoprecipitated with 14-3-3η.** (a) Immunoblot of total lysate and Na$_v$1.5-specific immunoprecipitated fraction of tsA201 cells 48 hours after transient co-expression of an empty vector, 14-3-3σ, 14-3-3η and Na$_v$1.5. Immunoprecipitation was performed with sepharose beads pre-coupled with anti-Na$_v$1.5 antibody. (b) Immunoblot of the control experiment testing aspecific binding of 14-3-3η to sepharose beads pre-coupled with immunoglobulin G (IgG) and anti-Na$_v$1.5 antibody. HA-tagged 14-3-3σ and 14-3-3η were revealed with an anti-HA antibody. Endogenous α1-syntrophin was used as a positive control for co-immunoprecipitation with Na$_v$1.5, and α-actin as a negative control. Yellow triangle indicates on the band corresponding to the co-immunoprecipitated 14-3-3η.

control experiment with sepharose beads on protein lysates from tsA201 cells lacking Na$_v$1.5 expression also revealed aspecific bands corresponding to 14-3-3η in immunoprecipitated fractions with both immunoglobulin G coupled beads and anti-Na$_v$1.5 antibody coupled beads (Fig 1B). Therefore, these results should be interpreted with high precaution. In conclusion, due to the overall weakness of Na$_v$1.5-immunoprecipitation with 14-3-3η protein, we suggest that if the interaction between Na$_v$1.5 and 14-3-3η genuinely occurs it is rather weak and/or transient.

## 14-3-3 proteins do not affect expression level of Na$_v$1.5 at the cell surface

Next, we assessed the role of 14-3-3 proteins on the cell surface density of Na$_v$1.5. This was motivated by findings from Pohl *et al.*, who suggested that 14-3-3 proteins modulate cell surface expression of membrane receptors and ion channels upon binding with the E3 ubiquitin ligase Nedd4-2 [16]. Specifically, they showed that the high affinity of 14-3-3 with phosphorylated Ser342 and Ser448 sites of Nedd4-2 induces its structural rearrangement with plausible exposure of its tryptophan-rich WW-domain [16]. Our group previously reported the interaction between the WW-domain of Nedd4-2 with the PY-motif of Na$_v$1.5 and its importance for Na$_v$1.5 cell surface localization [17,18]. According to our current results, the levels of the total and cell surface Na$_v$1.5 expression, represented by the biotinylated fraction, did not change upon overexpression of each individual 14-3-3 isoform when compared to the sodium channel alone (Fig 2A and 2B).

Of note, in tsA201, we detected endogenous mRNAs of all 14-3-3 isoforms (S3 Fig). In line with our results, transcripts of all 14-3-3 isoforms were previously reported for HEK293, the cell line from which tsA201 was originally derived [19]. Therefore, to inhibit endogenous 14-3-3/ligand interactions, we used difopein, a dimer of R18, the unphosphorylated peptide antagonist of 14-3-3 [7]. Difopein strongly interacts with two 14-3-3 proteins, stabilizing them

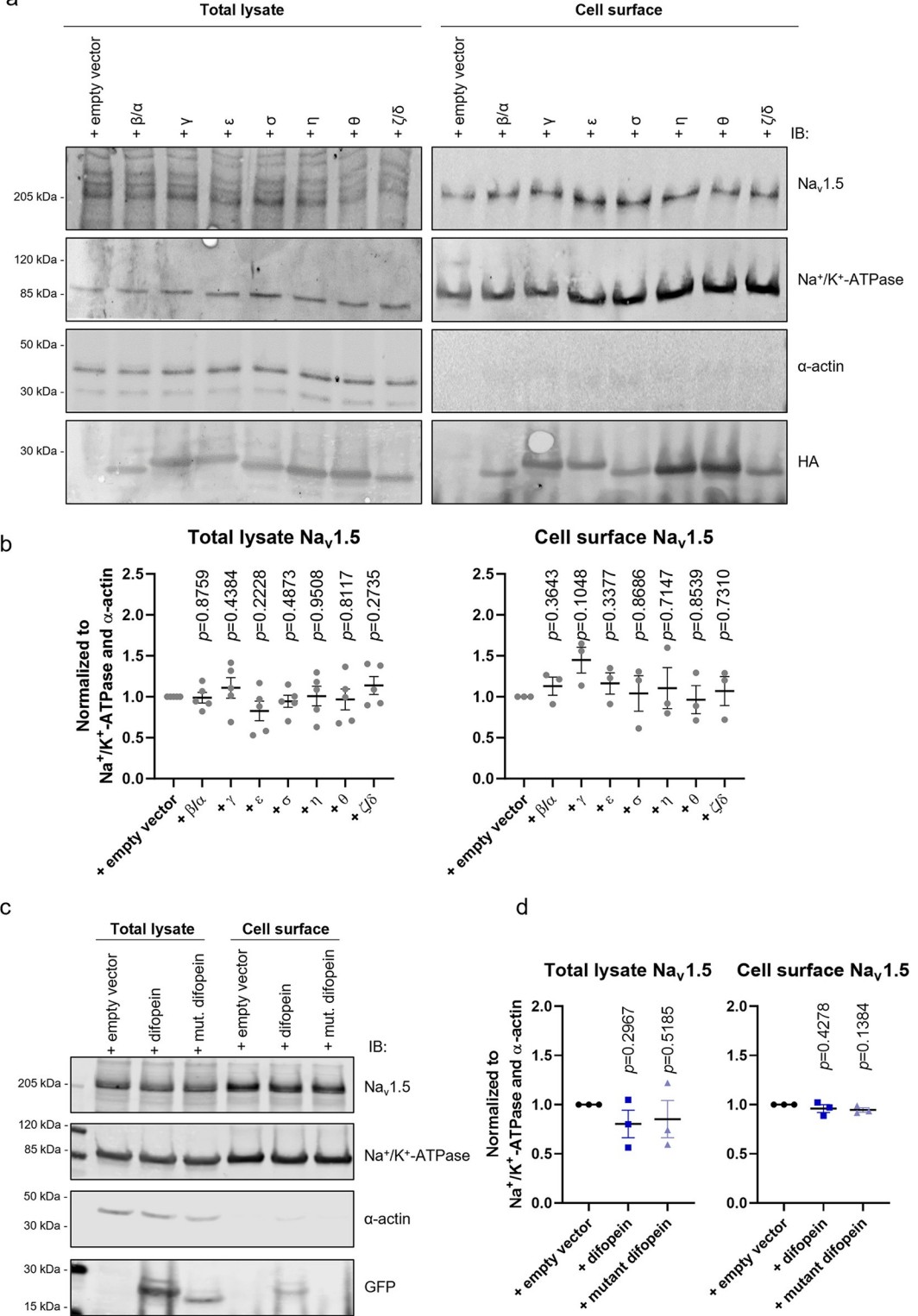

**Fig 2. 14-3-3 proteins do not affect the total expression level of Na$_v$1.5 and cell surface density.** Representative immunoblots of the total lysate and cell surface fraction of tsA201 stably expressing Na$_v$1.5 after 48 hours of transient overexpression of: (a) HA-tagged 14-3-3 proteins; (c) YFP-tagged difopein and its mutant. Endogenous Na$^+$/K$^+$-ATPase was used as a positive control of cell surface fraction, while α-actin was used as a negative control. (b and d) Intensities of Na$_v$1.5 in total protein lysate and at the cell surface were normalized to Na$^+$/K$^+$-ATPase and α-actin and to the control condition ("+ empty vector"). Data are presented as mean ± SEM from three-five biological replicates. Individual *p*-values, calculated with one-sample two-tailed t-test with hypothetical mean value = 1, are indicated in each panel.

in dimeric conformation, occupying their binding grooves, and preventing their interaction with other ligands [7]. We performed co-immunoprecipitation analysis to validate that difopein is an effective competitor for 14-3-3/ligand interactions in our heterologous expression system. First, we confirmed that 14-3-3 strongly interacts with difopein but not with its mutant (S4A Fig). As expected, difopein-YFP exhibited higher molecular weight than the mutant difopein-YFP, represented by the monomeric R18-D12K,E14K peptide (S4A Fig). Further, we demonstrated that 14-3-3 dimers were stabilized by difopein but not by its mutant (S4B Fig). Last, we confirmed that such an effect did not depend on any specific 14-3-3 isoform (SB Fig). Of note, difopein and its mutant did not affect the total and cell surface expression of Na$_v$1.5 (Fig 2C and 2D).

Therefore, we concluded that 14-3-3 proteins do not regulate Na$_v$1.5 expression and cell surface density in the tested heterologous expression system.

## Impact of 14-3-3 proteins onto Na$_v$1.5-mediated sodium current

Since of all 14-3-3 isoforms, only 14-3-3η weakly but reproducibly interacted with Na$_v$1.5 (Fig 1, S1 and S2 Figs), we assessed the whole cell recordings of the sodium current ($I_{Na}$) when transiently overexpressing *YWHAH*, encoding 14-3-3η. Moreover, to establish comparable conditions with the previous study that described a 14-3-3η-induced negative shift in $I_{Na}$ inactivation curve, we similarly co-expressed *SCN1B*, encoding the Na$_v$β1 subunit of the voltage-gated sodium channel [12]. In our heterologous expression system, neither 14-3-3η or Na$_v$β1 alone nor their combination modified the baseline whole-cell properties of $I_{Na}$, as described by the current-voltage (*I-V*) relationship, peak current density, reversal potential ($V_{rev}$) and half-maximal potentials ($V_{1/2}$) of activation and inactivation curves (Fig 3, Table 4). Similarly, disruption of endogenous 14-3-3/ligand interactions with difopein did not modify basal current properties of Na$_v$1.5, besides the slight but statistically insignificant effects on the peak current density and SSI (Fig 4, Table 5).

In summary, the isoform non-specific antagonist of 14-3-3, difopein, and 14-3-3η, the only isoform interacting with Na$_v$1.5, left the channel function unmodified.

## Inhibition of 14-3-3 dimerization does not affect the interaction between α-subunits of Na$_v$1.5

One role of the 14-3-3 family of adapter proteins is to stabilize multiprotein complexes [2]. In particular, 14-3-3η was proposed to mediate the coupled gating of Na$_v$1.5 dimers by bridging two α-subunits through their DI-DII loops [14]. We evaluated whether the 14-3-3 antagonist, difopein, affected Na$_v$1.5 dimerization. Our co-immunoprecipitation analysis confirmed the existence of Na$_v$1.5-Na$_v$1.5 interactions in a heterologous expression system, but these were not affected by difopein overexpression (Fig 5A and 5B). To confirm this finding in a more physiological and dynamic environment, we performed a NanoBiT assay. In brief, N- and C-termini of Na$_v$1.5 were tagged with LgBiT and SmBiT, respectively, which are parts of the bright and stable Nanoluciferase. Accordingly, when two Na$_v$1.5 physically interact, LgBiT and SmBiT would reconstitute Nanoluciferase, subsequently leading to luminescence emission (Fig 5C). Even though intrinsic interaction between LgBiT and SmBiT is rather weak ($K_d$ = 190 μM) [20], all experiments were accompanied by a simultaneous co-expression of Na$_v$1.5-LgBiT with the non-interacting HaloTag-SmBiT as a readout of the background luminescence (Fig 5C). In live tsA201 cells, the level of interaction between two Na$_v$1.5 was not modified by difopein (Fig 5D–5G). Interestingly, when comparing various locations of LgBiT and SmBiT tags on Na$_v$1.5, the brightest luminescent signal was detected when N-N termini

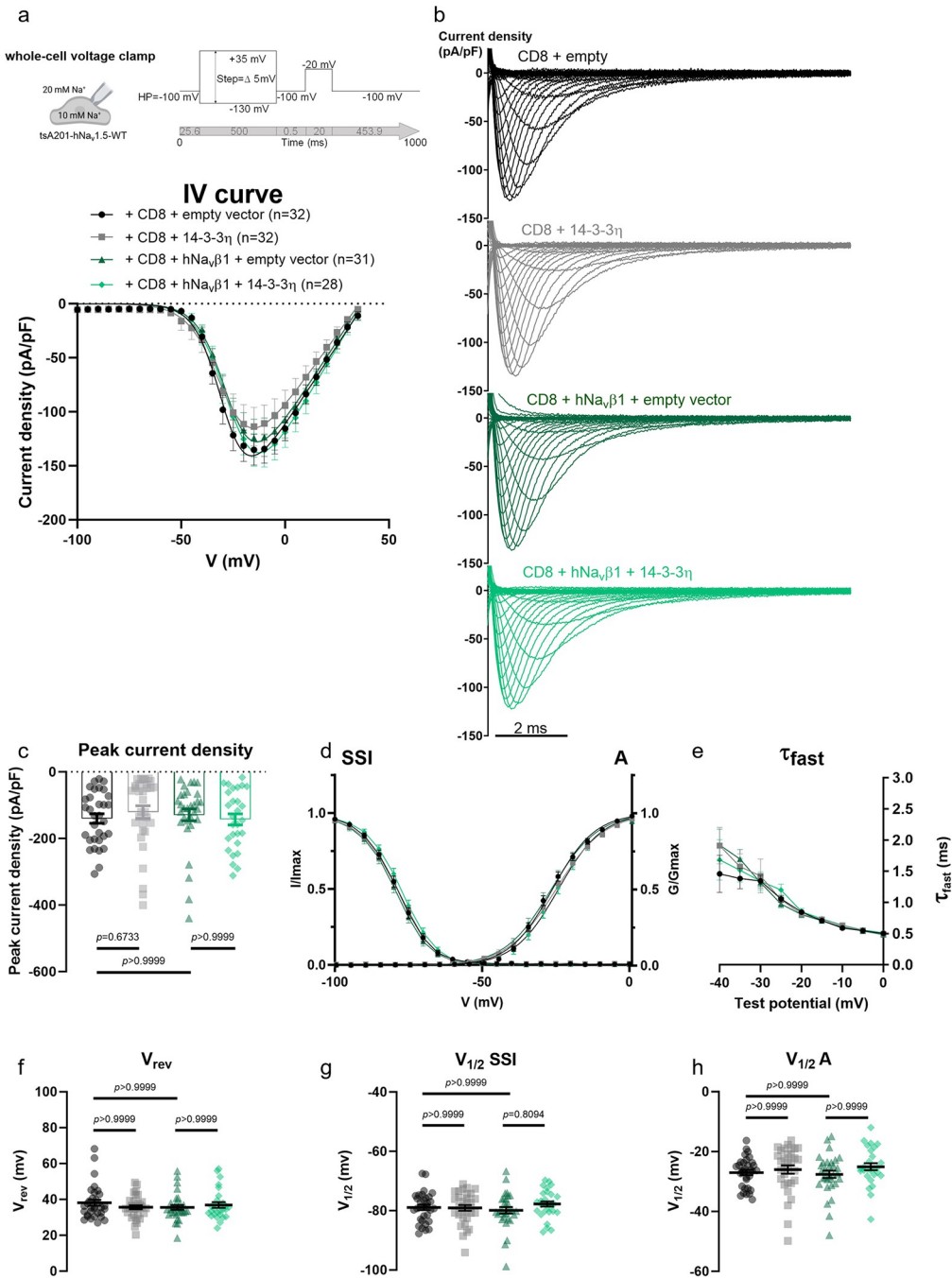

**Fig 3. 14-3-3η expressed alone or with Na$_v$β1 does not affect Na$_v$1.5-conducted sodium current. (a)** Current density-voltage (IV) relationships from tsA201-Na$_v$1.5 transiently expressing CD8 with empty vector, or 14-3-3η, and/ or Na$_v$β1. **(b)** Representative whole-cell $I_{Na}$ traces recorded from the listed conditions. **(c)** Peak current densities of each group. **(d)** Activation (A) and steady-state inactivation (SSI) curves. **(e)** Fast inactivation constant ($\tau_{fast}$) plotted as a function of test potential. **(f)** Reversal potential ($V_{rev}$). **(g)** Half-voltage of SSI ($V_{1/2}$ SSI). **(h)** Half-voltage of A ($V_{1/2}$ A). Values pertaining to the biophysical properties are shown in Table 4. Data are presented as mean ± SEM. *n*, number of cells taken for analysis. Kruskal-Wallis with post hoc Dunn's multiple comparisons has been performed in **(c, f, g,** and **h)** and the multiplicity adjusted *p*-values are indicated in each panel.

**Table 4. Biophysical properties of sodium currents conducted by $Na_v1.5$ stably present in tsA201 cells in combination with transiently expressed 14-3-3η and/or $Na_v\beta1$.**

| | Peak $I_{Na}$ density (pA/pF) | $V_{rev}$ (mV) | Activation | | Inactivation | | |
|---|---|---|---|---|---|---|---|
| | | | $k$ slope | $V_{1/2}$ (mV) | $k$ slope | $V_{1/2}$ (mV) | $\tau_{fast}$ (ms) at -15 mV |
| + CD8 + empty vector (n = 32) | -140.18 ± 14.11 | 38.15 ± 1.68 | 6.38 ± 0.17 | -27.01 ± 0.89 | 5.24 ± 0.12 | -78.93 ± 0.90 | 0.708 ± 0.02 |
| + CD8 + 14-3-3η (n = 32) | -120.69 ± 19.06 | 35.62 ± 1.13 | 6.45 ± 0.18 | -26.02 ± 1.38 | 5.22 ± 0.16 | -79.07 ± 0.95 | 0.727 ± 0.03 |
| + CD8 + hNa$_v\beta1$ + empty vector (n = 31) | -129.51 ± 17.83 | 35.55 ± 1.36 | 6.49 ± 0.13 | -27.61 ± 1.22 | 4.97 ± 0.14 | -79.86 ± 1.09 | 0.705 ± 0.03 |
| + CD8 + hNa$_v\beta1$ + 14-3-3η (n = 28) | -143.15 ± 16.60 | 36.88 ± 1.57 | 6.39 ± 0.17 | -25.05 ± 1.18 | 5.34 ± 0.17 | -77.75 ± 0.90 | 0.736 ± 0.03 |

CD8 was co-expressed to visualize successfully transfected cells. Peak sodium current ($I_{Na}$) densities are extracted from the respective I-V curves. Data are presented as mean ± SEM. n, number of cells analyzed.

were combined, compared to the N-C, C-N, and C-C conditions (Fig 5D–5G). It might indicate a preferred conformational orientation of α-subunits within $Na_v1.5$ dimers.

## Discussion

We reported that amongst seven 14-3-3 isoforms, only η (encoded by *YWHAH* gene) weakly co-immunoprecipitated with the major cardiac voltage-gated sodium channel $Na_v1.5$ when heterologously expressed in tsA201 cells. Nevertheless, neither 14-3-3η overexpression nor its inhibition with difopein affected the total protein level of $Na_v1.5$, its cell surface localization, dimerization, and basal biophysical properties of $I_{Na}$.

Our finding that 14-3-3η interacted with $Na_v1.5$ is in line with previous report of Allouis *et al.*, who demonstrated that 14-3-3 interacted with $Na_v1.5$ in Cos-7 and native mouse cardiac tissue and identified by yeast 2-hybrid screen DI-DII loop as a specific binding region of 14-3-3η [12]. Of note, when all cardiac voltage-gated sodium channels, including $Na_v1.5$, were immunoprecipitated from the mouse left ventricles using pan-$Na_v$ antibody, the most abundant peptides in the fraction were corresponding to 14-3-3ζ/δ, 14-3-3ε, 14-3-3γ, 14-3-3β/α, and 14-3-3σ, but not to 14-3-3η [21].

Furthermore, we reported that difopein and 14-3-3η did not modify $I_{Na}$ density and gating kinetics. Likewise, $I_{Na}$ density was not modified by exogenous 14-3-3η [12], by inhibition of endogenous 14-3-3η [22], and by isoform-unspecific antagonist, difopein [23]. However, regarding the activation and inactivation kinetics of $Na_v1.5$, several discrepancies were described. Similarly with our study, 14-3-3η overexpression did not modify conductance-voltage relationship [12]. But in contrast with our study, Zheng *et al.* demonstrated that difopein induced negative shift of the voltage-dependent conductance curve for the wild-type $Na_v1.5$ as well as for its truncated mutant Gly1642X with His558Arg polymorphism [23]. Furthermore, unlike our study, Allouis *et al.* showed that 14-3-3η shifted the inactivation curve to more negative potentials and decelerated the recovery from the inactivation of $Na_v1.5$ [12]. By co-expressing *SCN1B*, we eliminated the possibility that such discrepancy between our studies could be due to the presence of the $Na_v\beta1$ subunit. In fact, we did not observe any significant effects of *SCN1B* expression onto biophysical properties of the heterologous $Na_v1.5$. Indeed,

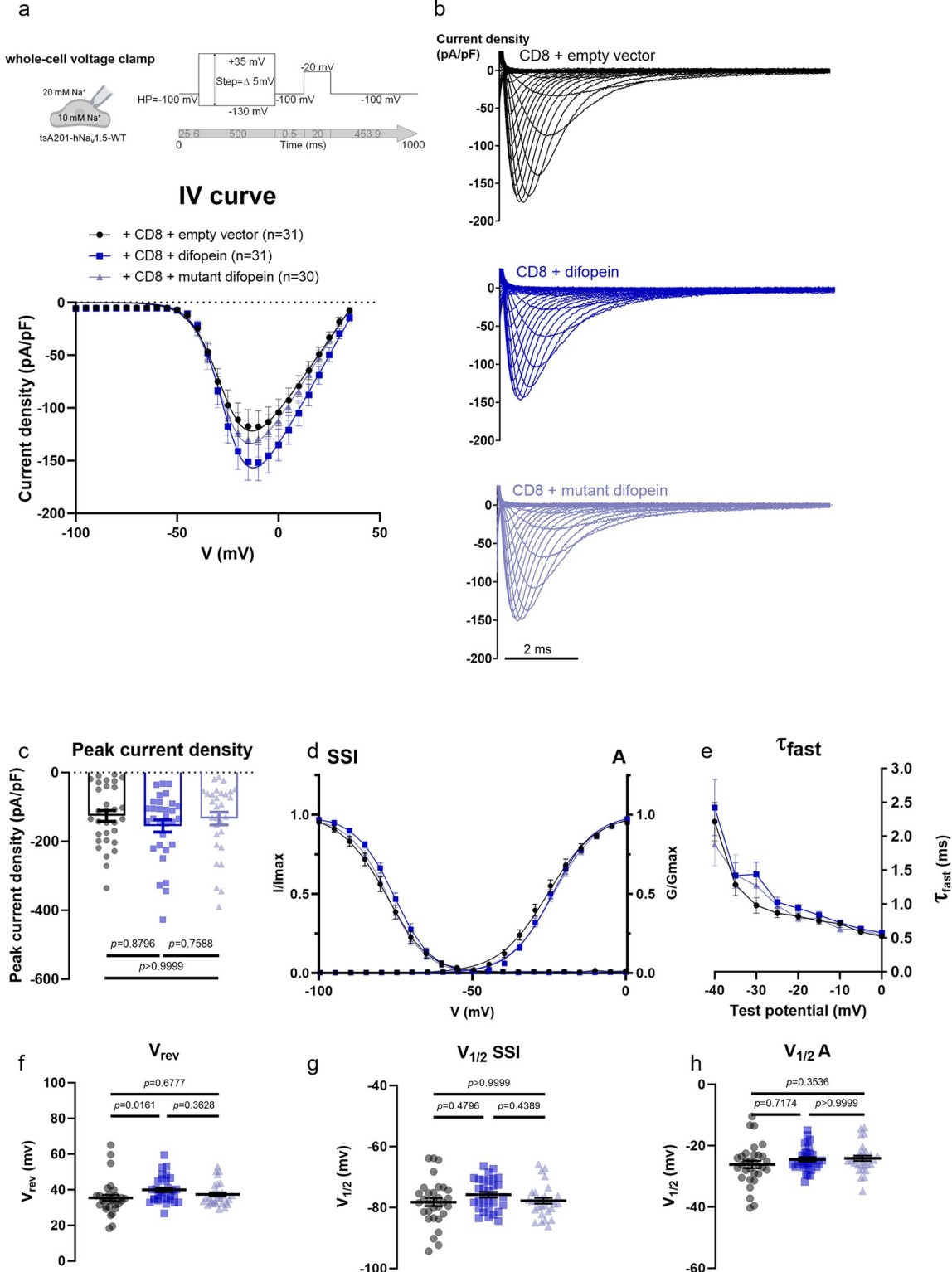

**Fig 4. Difopein, an antagonist of 14-3-3/ligand interactions, does not affect Na$_v$1.5-conducted sodium current.** (**a**) Current density-voltage (IV) relationships from Na$_v$1.5-expressing tsA201 cells transiently expressing CD8 with empty vector, difopein, or mutant of difopein. (**b**) Representative whole-cell $I_{Na}$ traces recorded from the listed conditions. (**c**) Peak current densities of each group. (**d**) Activation (A) and steady-state inactivation (SSI) curves. (**e**) Fast inactivation constant ($\tau_{fast}$) plotted as a function of test potential. (**f**) Reversal potential ($V_{rev}$). (**g**) Half-voltage of SSI ($V_{1/2}$ SSI). (**h**) Half-voltage of A ($V_{1/2}$ A). Values pertaining to the biophysical properties are shown in Table 5. Data are presented as mean ± SEM. *n*, number of cells taken for analysis. Kruskal-Wallis with post hoc Dunn's multiple comparisons has been performed in (**c, f, g**, and **h**) and the multiplicity adjusted *p*-values are indicated in each panel.

**Table 5. Biophysical properties of sodium currents conducted by Na$_v$1.5 stably present in tsA201 cells in combination with transiently expressed difopein and its mutant.**

| | Peak $I_{Na}$ density (pA/pF) | $V_{rev}$ (mV) | Activation | | Inactivation | | |
| --- | --- | --- | --- | --- | --- | --- | --- |
| | | | $k$ slope | $V_{1/2}$ (mV) | $k$ slope | $V_{1/2}$ (mV) | $\tau_{fast}$ (ms) at -15 mV |
| + CD8 + empty vector (n = 31) | 125.89 ± 15.65 | 35.47 ± 1.75 | 6.40 ± 0.14 | -26.14 ± 1.21 | 5.33 ± 0.17 | 78.22 ± 1.33 | 0.752 ± 0.04 |
| + CD8 + difopein (n = 31) | 155.07 ± 17.48 | 39.92 ± 1.30 | 6.30 ± 0.16 | -24.47 ± 0.72 | 5.30 ± 0.14 | 75.80 ± 0.92 | 0.835 ± 0.05 |
| + CD8 + mut. difopein (n = 30) | 133.26 ± 18.63 | 37.40 ± 1.17 | 6.51 ± 0.13 | -24.13 ± 0.84 | 5.55 ± 0.18 | 77.79 ± 0.94 | 0.787 ± 0.07 |

CD8 was co-expressed to visualize successfully transfected cells. Peak sodium current ($I_{Na}$) densities are extracted from the respective I-V curves. Data are presented as mean ± SEM. n, number of cells analyzed.

previous studies reported conflicting results regarding the impact of Na$_v$β1 on the function of Na$_v$1.5. While some groups detected changes of the peak current density and/or differences in the channel kinetics and gating properties, others like us did not observe any Na$_v$β1-mediated changes of $I_{Na}$ [24–31]. In particular, cardiomyocytes of *Scn1b* null mice exhibited increased macroscopic $I_{Na}$ without any differences in voltage dependence or kinetics of the current [29]. Further investigation revealed that this *Scn1b*-dependent $I_{Na}$ increase was mostly coming from the midsection of cardiomyocyte rather than the intercalated discs, it was tetrodotoxin-sensitive, and was concomitant with the increase of *Scn3a* mRNA [30]. Altogether suggesting on the unlikelihood of biophysical regulation of the native Na$_v$1.5 by Na$_v$β1 [30]. Another recent study demonstrated the importance of cellular model with its endogenous expression of Na$_v$β-subunits when investigating the biophysical properties of Na$_v$1.5 [31]. Specifically, co-expression of Na$_v$β1 did not exert any effect on Na$_v$1.5-mediated current in eHAP, human haploid fibroblasts-like cells with comparable endogenous mRNA levels of Na$_v$β-subunits to human embryonic kidney (HEK293) cells, but significantly affected $I_{Na}$ in gene-edited eHAP, where genes coding for endogenous *SCN1B-SCN4B* and their phylogenetic relatives were priorly disrupted [31]. Unlike the tsA201 cells used in our study, Allouis *et al.* used Cos-7 cells derived from monkey kidney fibroblasts as their heterologous expression system [12]. Like tsA201 cells, Cos-7 endogenously express all seven 14-3-3 isoforms, too [32]. However, their expression pattern, representing subcellular localization of 14-3-3 proteins, differed noticeably from HEK293 cells [32]. Furthermore, other protein partners that regulate the function of Na$_v$1.5 may also vary between cellular models and may explain the disparity of our outcomes.

Our observation that 14-3-3 proteins do not participate in the dimerization of heterologously expressed Na$_v$1.5 was also corroborated by previous reports [14,22]. Both studies highlighted that instead of directly mediating the formation of Na$_v$1.5 homo- and heterocomplexes with other ion channels (e.g., Na$_v$1.5-Na$_v$1.5 and Na$_v$1.5-K$_{ir2.1}$), 14-3-3η played a significant role in their biophysical coupling [14,22].

Zheng *et al.* also suggested additional indirect roles for 14-3-3-dependent regulation of Na$_v$1.5 [23]. Indeed, since Na$_v$1.5 is regulated by many proteins, including kinases, of which many are 14-3-3 substrates, too [33,34], it is plausible that 14-3-3 proteins could alter the function of Na$_v$1.5 indirectly through other partner proteins. Indeed, monomeric peptide R18, of which difopein is composed, disrupts 14-3-3γ/ε from protein kinase A (PKA), unmasking PKA's catalytic activity and promoting PKA-mediated phosphorylation of its downstream targets [35]. Na$_v$1.5 is among PKA's targets, and PKA stimulation in HEK293 cells has been shown to enhance $I_{Na}$ [36]. Furthermore, 14-3-3γ was shown to slow dephosphorylation of

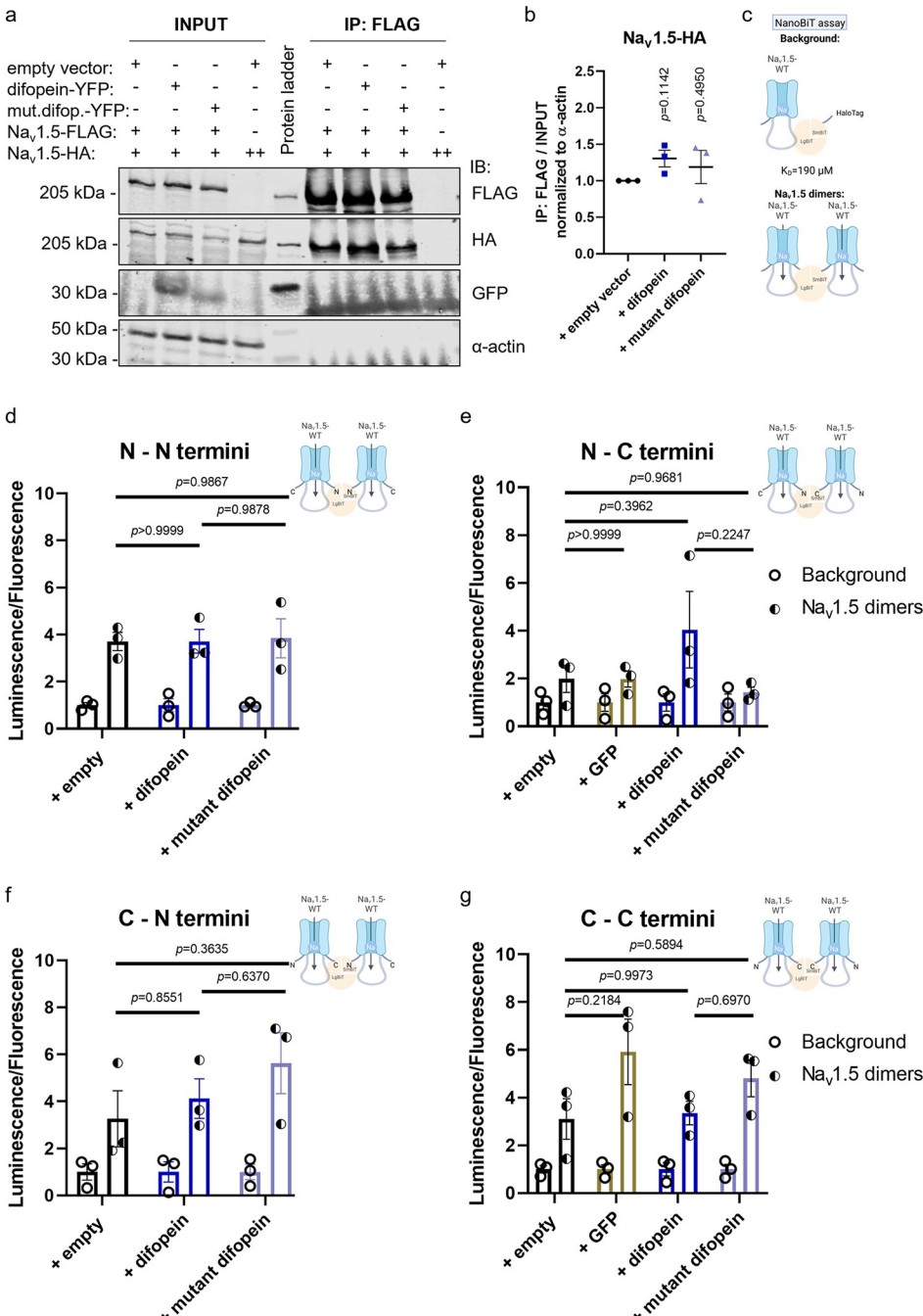

**Fig 5. Inhibition of 14-3-3/ligand interaction does not affect oligomerization between α-subunits of Na$_v$1.5.**
Difopein and its mutant did not affect co-immunoprecipitation between α subunits of Na$_v$1.5. **(a)** Representative
immunoblot of the total lysate ("INPUT") and FLAG-specific immunoprecipitated fraction ("IP: FLAG") of tsA201
cells transiently expressing Na$_v$1.5 with empty vector, difopein, or mutant of difopein. Overexpressed YFP-tagged
difopein and its mutant were revealed with an anti-GFP antibody. Endogenous α-actin was used as a negative control
for co-immunoprecipitation with Na$_v$1.5. **(b)** Intensity of co-immunoprecipitated Na$_v$1.5-HA was normalized to the
intensity of the total Na$_v$1.5-HA divided by the intensity of α-actin. Data are presented as mean ± SEM from three
biological replicates and are normalized to the control condition ("+ empty vector"). Individual $p$-values, calculated
with one-sample two-tailed t-test with hypothetical mean value = 1, are indicated in the panel. **(c)** Schematic
illustration of NanoBiT assay. Background signal was determined by co-expression of N- or C-termini LgBiT-Na$_v$1.5
with non-interacting control, SmBiT-HaloTag. **(d, e, f,** and **g)** In living cells difopein did not modify the level of Na$_v$1.5
dimers. Results of the NanoBiT assay are presented as relative intensity of luminescence (indicating the level of
protein-protein interactions) normalized to fluorescence (indicating the number of cells) in tsA201 cells transiently

transfected with *SCN5A* constructs. Each dataset was normalized to its relevant background. Different variations between Na$_v$1.5 N- and C-termini interactions were tested. Since difopein and its mutant were tagged with YFP, we also included control by co-transfection with GFP. NT, non-transfected. Data are presented as mean ± SEM from three biological replicates. The multiplicity adjusted *p*-values, calculated with one-way ANOVA and post hoc Tukey's multiple comparisons, are indicated in each panel.

calcium/calmodulin-dependent protein kinase (CaMKII) [37]. CaMKII is reported to mediate both gain- and loss-of-function effects of $I_{Na}$, highlighting its complex relationship with Na$_v$1.5 [33]. Moreover, 14-3-3ζ has been shown to regulate protein kinase C (PKC), which, in turn, also modulates Na$_v$1.5 activity through phosphorylation [38]. Nevertheless, in our heterologous expression system the effect of isoform-unspecific antagonist, difopein, on the regulation of Na$_v$1.5 was not evident.

In summary, ubiquitous presence of homo- and hetero-oligomerizing 14-3-3 proteins makes investigation of their direct role onto Na$_v$1.5 rather challenging. Furthermore, expression of 14-3-3 proteins and their downstream targets might drastically vary between different models, potentially providing for distinct direct and indirect modes of Na$_v$1.5 regulation.

## Limitations

To eliminate possible clonal specificity of our cell model, we developed polyclonal tsA201 stably expressing Na$_v$1.5. This led to high variability in whole-cell Na$_v$1.5 expression, which made single-cell electrophysiological analyses challenging, we strengthened the reliability of our data by testing a relatively large number of cells for each condition.

One of the limitations of this study is the presence of all endogenous 14-3-3 isoforms, including 14-3-3η, that could saturate Na$_v$1.5-binding sites and hence preclude interaction with the overexpressed proteins. To challenge this limitation, we compared endogenous and exogenous protein levels of 14-3-3η in our cell model. No noticeable level of endogenous 14-3-3η was detected when comparing it to the level of heterologously overexpressed 14-3-3η-HA (S5 Fig). Thus, we suggest that the endogenous level of 14-3-3η is rather low and is unlikely to compete with the overexpressed 14-3-3η-HA, especially since Na$_v$1.5 has also been overexpressed in our study. In addition to homodimerization 14-3-3 isoforms are also capable to form heterodimers [2,39]. Thus, it is possible that even when overexpressed 14-3-3η-comprised heterodimers may interact with substrates other than Na$_v$1.5, hindering its direct role on the channel. Therefore, using gene-editing techniques to silence specific isoforms would be advantageous to unravel the complexity of 14-3-3 signaling.

Another limitation of heterologously expressed proteins is the difficulty in controlling post-translational modifications, such as phosphorylation, that impact Na$_v$1.5 function and 14-3-3 signaling. A recent study reported a successful semisynthetic approach to stabilize Na$_v$1.5 phosphorylation in vitro [40]. Thus, it would be interesting to test 14-3-3-specific regulation under stably phosphorylated Na$_v$1.5.

While 14-3-3 might not act directly onto Na$_v$1.5, it was shown to act onto its macromolecular complex with other protein partners (e.g., PKA, PKC, CaMKII) and cardiac ion channels (e.g., Kir2.1) [16,22]. Because of the vast spectrum of 14-3-3 targets, one should be aware of the possible indirect effects measured in any experiment targeting 14-3-3 function. As such, any results obtained with difopein should be interpreted carefully.

In summary, while direct effects of 14-3-3, specifically isoform η, on the cardiac sodium channel are not evident in the tested heterologous expression system, it still may regulate Na$_v$1.5 through proteins of its macromolecular complex.

## Conclusions

We confirmed that 14-3-3η forms a macromolecular complex with human Na$_v$1.5 in a heterologous expression system. Although their interaction is rather weak and/or transient. Difopein is a potent tool to antagonize 14-3-3-ligand interactions but not the direct physical interaction between two Na$_v$1.5 α-subunits. Overexpression of 14-3-3η and difopein did not affect $I_{Na}$ density, gating kinetics as well as total and cell surface expression of Na$_v$1.5. Overall, the role of 14-3-3 proteins on the functionality of Na$_v$1.5 is dispensable in tsA201 cell line.

## Supporting information

**S1 Fig. Co-immunoprecipitation between Na$_v$1.5 and 14-3-3 isoforms.** 48 hours after transient overexpression of 14-3-3 β/α (encoded by *YWHAB*), 14-3-3 γ (encoded by *YWHAG*), 14-3-3 ε (encoded by *YWHAE*), 14-3-3 σ (encoded by *SFN*) in tsA201 expressing Na$_v$1.5 or Na$_v$1.5-FLAG. Endogenous α-1-syntrophin was used as a positive control for co-immunoprecipitation with Na$_v$1.5, and α-actin as a negative control.
(TIF)

**S2 Fig. Co-immunoprecipitation between Nav1.5 and 14-3-3 isoforms.** 48 hours after transient overexpression of 14-3-3η (encoded by *YWHAH*), 14-3-3θ (encoded by *YWHAQ*) and 14-3-3ζ/δ (encoded by *YWHAZ*) in tsA201 expressing Na$_v$1.5 or Nav1.5-FLAG. Endogenous α-1-syntrophin was used as a positive control for co-immunoprecipitation with Na$_v$1.5, and α-actin as a negative control. Intensity of Na$_v$1.5-immunoprecipitated 14-3-3η-HA was normalized to the intensity of the total 14-3-3η-HA divided by the intensity of α-actin. Data are presented as mean ± SEM from three biological replicates and are normalized to the control condition ("+ Na$_v$1.5"). Individual *p*-value, calculated with one-sample two-tailed t-test with hypothetical mean value = 1, is indicated in the panel.
(TIF)

**S3 Fig. Endogenous expression of 14-3-3 isoforms in tsA201 cell line.** Conventional RT-PCR on tsA201 WT cells showing endogenous transcripts of all seven 14-3-3 isoforms (14-3-3β/α encoded by *YWHAB*, 14-3-3γ encoded by *YWHAG*, 14-3-3ε encoded by *YWHAE*, 14-3-3σ encoded by *SFN*, 14-3-3η encoded by *YWHAH*, 14-3-3θ encoded by *YWHAQ*, and 14-3-3ζ/δ encoded by *YWHAZ*) as well as housekeeping genes (glyceraldehyde-3-phosphate dehydrogenase encoded by *GAPDH*, TATA-binding protein encoded by *TBP*, and proteasome 20S subunit β4 encoded by *PSMB4*) with their expected amplicon sizes.
(TIF)

**S4 Fig. Validation of difopein as an efficient competitor for 14-3-3/ligand interaction.** Co-immunoprecipitation analysis was performed 48 hours after transient overexpression of 14-3-3 in tsA201 WT cells. **(a)** Difopein strongly co-immunoprecipitated with 14-3-3 proteins, while the mutant of difopein did not. YFP-tagged difopein and its mutant were revealed with an anti-GFP antibody. **(b)** Difopein stabilized 14-3-3 dimers occupying its binding grooves, preventing 14-3-3 interactions with other ligands. The mutant of difopein did not stabilize 14-3-3 dimers and hence could be used as the closest relevant control for experiments involving difopein.
(TIF)

**S5 Fig. Investigation of endogenous 14-3-3η protein level in tsA201.** Immunoblotting analysis was performed 48 hours after transient overexpression of an empty vector or 14-3-3-HA proteins in tsA201 WT cells. Anti-14-3-3η antibody specifically detected overexpressed 14-3-3η and to a much lower extent 14-3-3ε. However, no noticeable level of endogenous 14-3-3η

was detected in tsA201 transfected with an empty vector. The successful overexpression of all seven mammalian 14-3-3 isoforms was revealed with an anti-HA antibody. Ponceau S staining was performed to verify equal protein loading.
(TIF)

## Acknowledgments

We thank Dr Zoja Selimi (University of Bern) and Prof Jan Kucera (University of Bern) for their invaluable insights and fruitful discussions. We thank Dr Sarah Vermij for proofreading of the article.

## Author Contributions

**Conceptualization:** Oksana Iamshanova, Daniela Ross-Kaschitza, Jean-Sébastien Rougier, Hugues Abriel.

**Data curation:** Oksana Iamshanova, Anne-Flore Hämmerli, Jean-Sébastien Rougier.

**Formal analysis:** Oksana Iamshanova, Anne-Flore Hämmerli.

**Funding acquisition:** Hugues Abriel.

**Investigation:** Oksana Iamshanova, Anne-Flore Hämmerli, Elise Ramaye, Arbresh Seljmani, Daniela Ross-Kaschitza, Noëlia Schärz, Maria Essers, Sabrina Guichard, Jean-Sébastien Rougier.

**Methodology:** Oksana Iamshanova.

**Visualization:** Oksana Iamshanova, Anne-Flore Hämmerli.

**Writing – original draft:** Oksana Iamshanova.

**Writing – review & editing:** Oksana Iamshanova, Jean-Sébastien Rougier, Hugues Abriel.

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
