## [Decision Letter · Decision Letter 0]

24 Aug 2023

PONE-D-23-21712The dispensability of 14-3-3 proteins for the regulation of human cardiac sodium channel Nav1.5PLOS ONE

Dear Dr. Iamshanova,

Thank you for submitting your manuscript to PLOS ONE, which has now been evaluated by three independent referees. All reviewers show interests in this study, but also raised some critical concerns that need to be addressed. Therefore, we feel that it has merit but does not fully meet PLOS ONE’s publication criteria as it currently stands, and we invite you to submit a revised version of the manuscript that addresses the points raised during the review process.

We look forward to receiving your revised manuscript.

Kind regards,

Zhiming Li, Ph.D.

Academic Editor

PLOS ONE

Journal Requirements:

2. Thank you for submitting the above manuscript to PLOS ONE. During our internal evaluation of the manuscript, we found significant text overlap between your submission and previous work in the [introduction, conclusion, etc.].

Please revise the manuscript to rephrase the duplicated text, cite your sources, and provide details as to how the current manuscript advances on previous work. Please note that further consideration is dependent on the submission of a manuscript that addresses these concerns about the overlap in text with published work.

[If the overlap is with the authors’ own works: Moreover, upon submission, authors must confirm that the manuscript, or any related manuscript, is not currently under consideration or accepted elsewhere. If related work has been submitted to PLOS ONE or elsewhere, authors must include a copy with the submitted article. Reviewers will be asked to comment on the overlap between related submissions (http://journals.plos.org/plosone/s/submission-guidelines#loc-related-manuscripts).]

We will carefully review your manuscript upon resubmission and further consideration of the manuscript is dependent on the text overlap being addressed in full. Please ensure that your revision is thorough as failure to address the concerns to our satisfaction may result in your submission not being considered further.

Reviewers' comments:

Reviewer's Responses to Questions

**Comments to the Author**

1. Is the manuscript technically sound, and do the data support the conclusions?

Reviewer #1: Yes

Reviewer #2: Partly

Reviewer #3: Partly

2. Has the statistical analysis been performed appropriately and rigorously? 

Reviewer #1: Yes

Reviewer #2: Yes

Reviewer #3: Yes

3. Have the authors made all data underlying the findings in their manuscript fully available?

Reviewer #1: Yes

Reviewer #2: Yes

Reviewer #3: Yes

4. Is the manuscript presented in an intelligible fashion and written in standard English?

Reviewer #1: Yes

Reviewer #2: Yes

Reviewer #3: Yes

5. Review Comments to the Author

Reviewer #1: Iamshanova et al. investigate Nav1.5 regulation by 14-3-3 proteins in tsA201 cells. Endogenous mRNAs for all seven 14-3-3 isoforms were detectable in this cell line. Out of those, only one, 14-3-3η, was found to reliably and reproducibly interact with Nav1.5 in co-immunoprecipitation experiments. However, this interaction appeared to be weak and/or transient. Further, analysis of immunoblots showed that both total and cell surface Nav1.5 expression remained unchanged upon overexpression of each individual 14-3-3 isoform compared to the sodium channel alone. Subsequently, difopein and a difopein mutant were used as competitors for 14-3-3/ligand interactions. Co-immunoprecipitation experiments showed 14-3-3 dimers strongly interacting with and being stabilized by difopein but not its mutant, independent of 14-3-3 isoform. Neither difopein nor its mutant affected total or cell surface Nav1.5 expression. To investigate any sodium current effects, 14-3-3η was overexpressed with and without co-expressed SCN1B and whole-cell patch clamp data was collected. Neither 14-3-3η nor SCN1B, alone or in combination, were found to modify baseline whole-cell properties of the sodium current. Difopein also failed to alter these properties. Consistent with previous studies, further co-immunoprecipitation analysis confirmed the presence of Nav1.5-Nav1.5 interactions, but these were unaffected by the overexpression difopein. The researchers concluded that 14-3-3 proteins do not regulate Nav1.5 expression, cell surface density, or channel function in tsA201 cells.

This manuscript provides interesting data suggesting that, at least in tsA201 cells, 14-3-3 proteins do not regulate Nav1.5 to the extent previously described in older studies. Even if these results are due entirely to the differences in cell lines used, they can still provide important groundwork for further investigations into a potentially complex and expression-system-dependent regulatory mechanism for Nav1.5. Overall, the manuscript is well written and thorough, but still has some room for improvement, as mentioned in the following critiques.

Critiques:

1. In the “Impact of 14-3-3 proteins onto Nav1.5-mediated sodium current” section, it was found that expression of SCN1B by itself had no effect on the current properties of Nav1.5. However, no information is given regarding any potential presence of endogenous sodium channel beta subunits in this expression system. The lack of effect could be the result of endogenous beta subunits filling the role of SCN1B when it is not expressed.

2. Figure 4 appears to show slight but statistically insignificant effects on peak current and SSI. The authors may want to acknowledge this, while still specifying that these effects are not statistically significant. Such a small effect could be consistent with the very weak association seen between 14-3-3η and Nav1.5 in the co-immunoprecipitation experiments. Additionally, the representative traces chosen appear to show “CD8 + difopein” inactivating slightly faster than the other two conditions. If this is not generally the case, the authors may want to use alternative traces. Further, it may be worthwhile to include the inactivation time constants, as well as curves showing the recovery from inactivation.

3. In Figure 3, the representative trace matched by color to the IV curve for “CD8 + hNavB1 + empty vector” seems to be mislabeled as “CD8 + mutant difopein.”

Reviewer #2: Iamshanova et al have undertaken a combination of biochemical and biophysical analyses to investigate the interaction between the human Nav1.5 protein and the 14-3-3 proteins in a heterologous cell expression system. The rationale for assessing this interaction derives from prior work demonstrating a direct interaction between 14-3-3 eta and Nav1.5, and complementary work suggesting Nav1.5 is functionally regulated by 14-3-3 proteins. However, the latter study did not establish which isoform is responsible. In the present study the authors demonstrate that of the seven isoforms only 14-3-3 eta forms macromolecular complexes with Nav1.5. This corroborates the original study, however the relative pull down of 14-3-3 eta is still limited, as acknowledged by the authors. Further, the authors observe no functional consequence of this interaction.

MAJOR

There are some issues with respect to the quality of the Co-IP and western blots, and the corresponding explanations within the main body text. These are detailed below:

Fig 1.

The blots for the IP have very high background. In both the Nav1.5 and HA (14-3-3) blots there are strong smears running down the borders of all 3 lanes (this does not appear to be an issue for α1-syntrophin or α-actin). As such the band highlighted by the yellow triangle is faint relative to these smears.

The relative signal intensity of the two HA tagged 14-3-3 bands in the input fraction do not suggest equal expression (actin loading control suggests consistent loading). 14-3-3 eta has a visibly higher signal intensity than 14-3-3 sigma. In addition to this, it looks as though there may be two bands present in the input fraction. A dense lower band and a relatively faint upper band (at least for 14-3-3 eta). Does the larger molecular weight band correspond to the immunoprecipitated fraction? If so, does the relatively weak signal in 14-3-3 sigma potentially preclude any successful IP (does a longer exposure or increased protein loading overcome this?). Do the authors have any suggestion as to the identity of the two migratory bands and why only one, notably the weaker of the two, is precipitated.

The identity of the bands in the Nav1.5 blots is unclear. The input lanes show a predominant band approximating 205 kDa (smaller than the molecular weight of Nav1.5) together with a faint larger migratory band. In the IP fraction the 205 kDa band is most abundant and an extra intermediate molecular weight band is also present (3 bands in total). What do the different migratory forms correspond to? (These multiple bands are not present in supplementary figs S1 or S2). Which have been included in the total Nav1.5 quantification?

Are you sure that 14-3-3 η is not interacting with the Sepharose-anti-Nav1.5 coupled beads directly? (similar to what is being seen with the FLAG beads in figs S1 and S2?)

Fig 2. There are now four or five bands present for Nav1.5 in the total lysate in panel a (there do not appear to be as many in panel c), while only one is present at the cell surface (the lowest migratory band).

Lines 243-246. ‘From all the isoforms, only 14-3-3η exhibited reproducibly brighter band intensity in the FLAG-immunoprecipitated fraction compared to the non-tagged fraction; hence, we concluded that only this isoform reliably interacts with Nav1.5’

There should not be any 14-3-3 protein in the immunoprecipitated fraction in the absence of a FLAG tagged protein. To suggest the relative intensity is important here is dubious given the relative intensity of the input fractions where 14-3-3 η is the most abundant. Additionally, it is not clear what has been normalised in the graph in fig S2, or how this has been achieved. The signal of 14-3-3 η bands in the input fraction is almost certainly saturated.

Clarify ‘thoroughly washing the beads with TBS’ particularly given the non-specific interactions of 14-3-3 with the FLAG beads in the supplementary figures.

I am not convinced that the appropriate statistical analysis has been carried out for the data in figure 2. I appreciate in the methods it is stated that when the data is normalised to the control condition a t-test has been used. But all 7 different experimental samples were run together with the single control, in which you were quantifying the same protein and presumably in a linear range for densitometric analysis. So, wouldn’t a Kruskal-Wallis ANOVA (non-parametric) be more appropriate?

Lines 291-293. Please clarify. In fig S4a the different migratory bands are not clear. What are the different molecular weights? Also, the subsequent sentence states difopein stabilises 14-3-3 dimers. The monomer has a molecular weight of 30kDa, so presumably the dimer is 60kDa? Not only is this not made clear in the text, fig S4b is not labelled to highlight monomeric/dimeric bands, or even show any migration around 60 kDa. The panels display a region covering two protein ladder markers of 25 kDa and 30 kDa.

It is unclear why these supplementary experiments did not include the only isoform (14-3-3 eta) that has been demonstrated to interact with Nav1.5.

The whole-cell voltage clamp raw data are of high quality. A minor point, in Fig 4, the representative current waveforms in panels b for CD8 + empty vector do not appear to correspond with the IV curve. The last current waveform displayed peaks around -50 pA/pF, yet on the IV curve it is almost at 0 pA/pF. Are some currents missing from the representative plot in panel B, or does the voltage-dependence of this particular data set differ from that represented in panel A?

Although the raw data is of high quality, there is a noticeable variability in some key parameters, and an unusually high n number. Panel e, a significant difference (P=0.0161) is highlighted between CD8 + empty vector and CD8 + difopein. No explanation has been given for this variation in reversal potential. It is also peculiar that there is such a large scatter of the Vrev, particularly for CD8 + empty vector, with values ranging from 20 mV up to 60 or 65 mV. Taking into account the reported internal and external Na+ concentrations in panel a, the equilibrium potential should of the order of 26 mV. Do the currents that strongly deviate (i.e. those around 60 mV) also have V1/2s that deviate from the mean? I note that the authors have commented on the polyclonal nature of their cell line as being implicated in the variability of Nav1.5 expression - this could account for variations such as current density, but not reversal potential.

Line 421 – 425. Why does co-expressing SCN1B eliminate the possibility that the discrepancy could be due to the presence of the NavB1 subunit? Is this referring to endogenous expression? Which experiments was SCN1B co-expressed in? The methods are not clear.

MINOR

In Fig 2, I think the 205 kDa label is in the wrong location on panel a. Additionally, the data in the graphs in panel b occupy less than 50% of the graphical space. Are 4 significant figures necessary for the P values presented?

Line 44 – grammatical error, either a comma or ‘the’ is missing between heart and voltage-gated.

Line 124 and Line 139 - Cell pellets or cell monolayers are referred to as ‘prewashed (PBS)’ what does this mean? Cells were washed 'x' times in PBS?

Line 149 – typo? ‘nProtein A Sepharose’

Lines 152 -154 Clarify ‘prewashed’ and ‘thoroughly washing’ (this wording occurs again at line 164, and also needs clarification).

Lines 366-370. The luminescent signal intensity in Fig 5 is of a similar order of magnitude for d, f and g. Only e (N - C) has a lower signal intensity. So the conclusion that N-N and C-C termini combinations were brightest compared to N-C and C-N is not entirely accurate.

Line 435 – 439. Ref 14 is Clatot et al. They did observe 14-3-3 participating in dimerization and did not specifically assess 14-3-3 η.

Reviewer #3: It is now fairly clear that sodium channels such as Nav1.5 can interact on the plasma membrane as dimers. In 2017, Clatot et al reported that 14-3-3 mediated this interaction. The current paper addresses the necesssity of 14-3-3 in this process. Although the authors show a weak interaction between 14-3-3ƞ (but no other 14-3-3 isoforms) and Nav1.5, they find no evidence that 14-3-3 is required for Nav1.5 dimerisation; neither can they find evidence for any functional requirement of 14-3-3 in Nav1.5 dimerisation or activity. If correct, this is important, as on the face of it, the data is not consistent with outher work on the same question.

The fundamental problem is that the cells used in these experiments express endogenous 14-3-3 isoform, including the ƞ isoform (see Sup Fig 3). This means that any signal generated by overexpressing 14-3-3 isoforms may be small. This may for example, explain whythe abslolute amount of 14-3-3ƞ immunoprecipitated with Nav1.5 (Fig 1) is weak. The authors use difopein (an inhibitor of 14-3-3 dimerisation) to show that 14-3-3 dimerisation does not affect Nav1.5 dimerisation. But it is possible that 14-3-3 is required for the initial assembly of the Nav1.5 dimers, rather than a continuous requirment. This is why knocking down 14-3-3ƞ by for example, RNAi, is required to really rule out an role in Nav1.5 dimerisation in this particular experimental system.

Some additional points:

Figure 1: the 14-3-3 band in the IP fraction (indicated by the yellow arrow head) has a noticable runsnoticably higher than the 14-3-3 protein in the input. But interestingly, the input band does have a fainter, higher band that could be the 14-3-3 in the IP fraction. Does this indicate a small fraction of modified (eg phosphorylation?) 14-3-3?

The authors say "..when comparing various locations of LgBiT and SmBiT tags on Nav1.5, the brightest luminescent signal was detected when N-N and C-C termini were combined, compared to the n-C and C-N conditions (Figs 5d-g)". This statement may be true fo N-C, but C-N doesn't look much different from N-N .

6. PLOS authors have the option to publish the peer review history of their article (what does this mean?). If published, this will include your full peer review and any attached files.

Reviewer #1: No

Reviewer #2: No

Reviewer #3: No

---

## [Author Response · Author response to Decision Letter 0]

29 Dec 2023

Our response exceeds 20,000 characters. Therefore, we have provided it as an attachment file named "Response to Reviewers".

---

## [Decision Letter · Decision Letter 1]

17 Jan 2024

PONE-D-23-21712R1The dispensability of 14-3-3 proteins for the regulation of human cardiac sodium channel Nav1.5PLOS ONE

Dear Dr. Iamshanova,

Thank you for submitting your revised manuscript to PLOS ONE. Both reviewers agree that the manuscript has been greatly improved. However, reviewer 2 still has some concerns about some of the data and conclusions, which I think should be addressed before we move forward. Therefore, we invite you to submit a revised version of the manuscript that addresses the remaining points raised during the review process.

We look forward to receiving your revised manuscript.

Kind regards,

Zhiming Li, Ph.D.

Academic Editor

PLOS ONE

Journal Requirements:

Reviewers' comments:

Reviewer's Responses to Questions

**Comments to the Author**

1. If the authors have adequately addressed your comments raised in a previous round of review and you feel that this manuscript is now acceptable for publication, you may indicate that here to bypass the “Comments to the Author” section, enter your conflict of interest statement in the “Confidential to Editor” section, and submit your "Accept" recommendation.

Reviewer #1: All comments have been addressed

Reviewer #2: (No Response)

2. Is the manuscript technically sound, and do the data support the conclusions?

Reviewer #1: Yes

Reviewer #2: Yes

3. Has the statistical analysis been performed appropriately and rigorously? 

Reviewer #1: Yes

Reviewer #2: Yes

4. Have the authors made all data underlying the findings in their manuscript fully available?

Reviewer #1: Yes

Reviewer #2: Yes

5. Is the manuscript presented in an intelligible fashion and written in standard English?

Reviewer #1: Yes

Reviewer #2: Yes

6. Review Comments to the Author

Reviewer #1: (No Response)

Reviewer #2: The revisions and added experiments strengthen the authors findings and sufficiently address my comments. I apologise for the confusion regarding the point of the role of 14-3-3 in Nav1.5 dimerization. The authors are correct 14-3-3 is not necessary for dimerization, as observed here and in prior studies, although it could possibly further stabilise the complex. In contrast with others, no functional effects were observed leading the authors to conclude that 14-3-3 proteins do not regulate Nav1.5 in tsa201 cells (the model system perhaps accounting for the variability).

Remaining points:

Lines 217 – 220: Description of double exponential function has presumably omitted the word ‘decay’ or ‘inactivation’ from line 218 (after current). NB. The values for tfast vary unexpectedly at -30 mV in fig4e for CD8 + difopein. The CD8 + empty vector values are surprisingly similar between -40 to -30 mV in fig 3e

Line 217 – V1/2 also applies to inactivation.

Line 442 – 445.

“Indeed, previous studies reported conflicting results regarding the impact of Navβ1 on the function of Nav1.5. While some groups detected changes of the peak current density and/or differences in the channel kinetics and gating properties, others like us did not observe any Navβ1-mediated changes of INa [24–28].”

The effects of beta 1 are variable in the literature, but of references 24 – 28, I do not agree that any of these support an absence of any effect of beta 1 on INa.

24: Increased current density, negative shift of V1/2 activation and accelerated recovery.

25: Authors have previously shown that beta 1 causes a negative shift of Nav1.5 SSI in both oocytes and HEK293 cells (HEK293 cells; supplementary data in Zhu et al 2017 JGP). In the referenced paper (25), WT and SCN1B null myocytes are compared and peak current is smaller in the presence of beta 1.

26: Shift of voltage dependence of SSI

27: Shift of voltage dependence of SSI

28: Shift of voltage dependence of SSI

Regarding the variation of vrev. I agree with the authors, the liquid junction potential likely accounts for the difference in recorded values of vrev when compared with the theoretical value, which is fine assuming it is stable. It is also highly unlikely that difopein is affecting the selective Na+ permeability of the channel pore. But, in some cells there is a large deviation of vrev. Assuming that the Na+ permeability is not affected, this variation has most likely arisen from some technical issue, for which the vector specificity and tags do not seem like a likely source.... a more likely culprit (though not necessarily the case here!) would be something like the integrity of the electrode (e.g need for re-chloridisation). If this, or some other source was identified it would be a reason to exclude the affected data points. In the absence of some identifiable reason, of course under the present exclusion criteria stated by the authors the data should be included, but perhaps commented on (assuming the authors find this variability odd too).

Regarding fig 2. I was not suggesting that you should remove the panels in b, just that the scale could be adjusted to fully visualise the data points. The values of your data all appear to lie in the region of ~0.5 – <2, but the scale of your y-axis extends to 4.

7. PLOS authors have the option to publish the peer review history of their article (what does this mean?). If published, this will include your full peer review and any attached files.

Reviewer #1: No

Reviewer #2: No

---

## [Author Response · Author response to Decision Letter 1]

24 Jan 2024

Manuscript PONE-D-23-21712R1

Response to Reviewers

Dear Dr. Li Zhiming,

Thank you for giving us the opportunity to submit the second revised version of the manuscript “The dispensability of 14-3-3 proteins for the regulation of human cardiac sodium channel Nav1.5” for publication in PLOS ONE. We appreciate the time and effort that you and reviewers dedicated to our manuscript and are grateful for the valuable comments and suggestions. We have accepted all the changes that were previously suggested by the reviewers and have now incorporated most of the suggestions made by Reviewer #2. Those changes are highlighted in the updated “Revised Manuscript with Track Changes 240124” file. Please see below, our point-by-point response to the reviewers’ comments. All line numbers refer to the updated “Revised Manuscript with Track Changes 240124” file.

Reviewer #2: The revisions and added experiments strengthen the authors findings and sufficiently address my comments. I apologise for the confusion regarding the point of the role of 14-3-3 in Nav1.5 dimerization. The authors are correct 14-3-3 is not necessary for dimerization, as observed here and in prior studies, although it could possibly further stabilise the complex. In contrast with others, no functional effects were observed leading the authors to conclude that 14-3-3 proteins do not regulate Nav1.5 in tsa201 cells (the model system perhaps accounting for the variability).

Response: Thank you! We greatly appreciate Reviewer’s openness and constructive feedback. 

Remaining points:

Lines 217 – 220: Description of double exponential function has presumably omitted the word ‘decay’ or ‘inactivation’ from line 218 (after current). NB. The values for tfast vary unexpectedly at -30 mV in fig4e for CD8 + difopein. The CD8 + empty vector values are surprisingly similar between -40 to -30 mV in fig 3e

Response: Thank you for spotting it! Changes were applied as follows.

Line 228: “.. were extracted from fitting onset of current decay ..”

Line 217 – V1/2 also applies to inactivation.

Lines 213-223: “.., where y is the normalized peak current (pA/pF) at a given holding potential, g is the maximal conductance, Vm is the membrane potential, Vrev is the reversal potential for Na+, V1/2 is the potential at which half of the channels are activated and k is the slope factor of activation. Activation (A) curves were fitted with the Boltzmann equation y = 1-(1/(1 + exp[(Vm – V1/2)/k])), where y is the normalized conductance at a given holding potential, Vm is the membrane potential, V1/2 is the potential at which half of the channels are activated and k is the slope factor of activation. SSI curves were fitted with the Boltzmann equation y = 1/(1 + exp[(Vm – V1/2)/k]), where y is the normalized current at a given holding potential, Vm is the membrane potential, V1/2 is the potential at which half of the channels inactivated and k is the slope factor of inactivation.”

Line 442 – 445.

“Indeed, previous studies reported conflicting results regarding the impact of Navβ1 on the function of Nav1.5. While some groups detected changes of the peak current density and/or differences in the channel kinetics and gating properties, others like us did not observe any Navβ1-mediated changes of INa [24–28].”

The effects of beta 1 are variable in the literature, but of references 24 – 28, I do not agree that any of these support an absence of any effect of beta 1 on INa.

24: Increased current density, negative shift of V1/2 activation and accelerated recovery.

25: Authors have previously shown that beta 1 causes a negative shift of Nav1.5 SSI in both oocytes and HEK293 cells (HEK293 cells; supplementary data in Zhu et al 2017 JGP). In the referenced paper (25), WT and SCN1B null myocytes are compared and peak current is smaller in the presence of beta 1.

26: Shift of voltage dependence of SSI

27: Shift of voltage dependence of SSI

28: Shift of voltage dependence of SSI

Response: Thank you for the valid comment! We have added more references into this paragraph and developed further our argumentation as follows. 

Lines 454-468: “..of INa [24–31]. In particular, cardiomyocytes of Scn1b null mice exhibited increased macroscopic INa without any differences in voltage dependence or kinetics of the current [29]. Further investigation revealed that this Scn1b-dependent INa increase was mostly coming from the midsection of cardiomyocyte rather than the intercalated discs, it was tetrodotoxin-sensitive, and was concomitant with the increase of Scn3a mRNA [30]. Altogether suggesting on the unlikelihood of biophysical regulation of the native Nav1.5 by Navβ1 [30]. Another recent study demonstrated the importance of cellular model with its endogenous expression of Navβ-subunits when investigating the biophysical properties of Nav1.5 [31]. Specifically, co-expression of Navβ1 did not exert any effect on Nav1.5-mediated current in eHAP, human haploid fibroblasts-like cells with comparable endogenous mRNA levels of Navβ-subunits to human embryonic kidney (HEK293) cells, but significantly affected INa in gene-edited eHAP, where genes coding for endogenous SCN1B-SCN4B and their phylogenetic relatives were priorly disrupted [31].”

Line 472: “.. from HEK293 cells [36].”

References added:

Lines 655-667:

“29. Lopez-Santiago LF, Meadows LS, Ernst SJ, Chen C, Malhotra JD, McEwen DP, et al. Sodium channel Scn1b null mice exhibit prolonged QT and RR intervals. J Mol Cell Cardiol. 2007;43: 636–647. doi:10.1016/j.yjmcc.2007.07.062

30. Lin X, O’Malley H, Chen C, Auerbach D, Foster M, Shekhar A, et al. Scn1b deletion leads to increased tetrodotoxin-sensitive sodium current, altered intracellular calcium homeostasis and arrhythmias in murine hearts. Journal of Physiology. 2015;593: 1389–1407. doi:10.1113/jphysiol.2014.277699

31. Minard AY, Clark CJ, Ahern CA, Piper RC. Beta-subunit-eliminated eHAP expression (BeHAPe) cells reveal subunit regulation of the cardiac voltage-gated sodium channel. Journal of Biological Chemistry. 2023;299. doi:10.1016/j.jbc.2023.105132”

Regarding the variation of vrev. I agree with the authors, the liquid junction potential likely accounts for the difference in recorded values of vrev when compared with the theoretical value, which is fine assuming it is stable. It is also highly unlikely that difopein is affecting the selective Na+ permeability of the channel pore. But, in some cells there is a large deviation of vrev. Assuming that the Na+ permeability is not affected, this variation has most likely arisen from some technical issue, for which the vector specificity and tags do not seem like a likely source.... a more likely culprit (though not necessarily the case here!) would be something like the integrity of the electrode (e.g need for re-chloridisation). If this, or some other source was identified it would be a reason to exclude the affected data points. In the absence of some identifiable reason, of course under the present exclusion criteria stated by the authors the data should be included, but perhaps commented on (assuming the authors find this variability odd too).

Response: Thank you for your valid comment! Indeed, we also found these 3 cells odd. However, when trying to identify the “culprit” of such effect based on the experimental dish, day or condition, we could not conclude on any obvious technical or biological effect that might cause such variability. 

Regarding fig 2. I was not suggesting that you should remove the panels in b, just that the scale could be adjusted to fully visualise the data points. The values of your data all appear to lie in the region of ~0.5 – <2, but the scale of your y-axis extends to 4.

Response: Thank you for the clarification! We have adapted the y axes in Fig. 2b from 0 to 2.5 as kindly suggested by the Reviewer.

---

## [Editor Report · Decision Letter 2]

31 Jan 2024

The dispensability of 14-3-3 proteins for the regulation of human cardiac sodium channel Nav1.5

PONE-D-23-21712R2

Dear Dr. Iamshanova,

Thank you for the efforts to address the reviewers' comments. We’re pleased to inform you that your manuscript has been judged scientifically suitable for publication and will be formally accepted for publication once it meets all outstanding technical requirements.

Kind regards,

Zhiming Li, Ph.D.

Academic Editor

PLOS ONE
---

## [Editor Report · Acceptance letter]

26 Feb 2024

PONE-D-23-21712R2 

PLOS ONE

Dear Dr. Iamshanova, 

I'm pleased to inform you that your manuscript has been deemed suitable for publication in PLOS ONE. Congratulations! Your manuscript is now being handed over to our production team.

Kind regards, 

on behalf of

Dr. Zhiming Li 

Academic Editor

PLOS ONE